# Hydrogen Peroxide Effects on Natural-Sourced Polysacchrides: Free Radical Formation/Production, Degradation Process, and Reaction Mechanism—A Critical Synopsis

**DOI:** 10.3390/foods10040699

**Published:** 2021-03-25

**Authors:** Chigozie E. Ofoedu, Lijun You, Chijioke M. Osuji, Jude O. Iwouno, Ngozi O. Kabuo, Moses Ojukwu, Ijeoma M. Agunwah, James S. Chacha, Onyinye P. Muobike, Adedoyin O. Agunbiade, Giacomo Sardo, Gioacchino Bono, Charles Odilichukwu R. Okpala, Małgorzata Korzeniowska

**Affiliations:** 1Department of Food Science and Technology, School of Engineering and Engineering Technology, Federal University of Technology, Owerri, 460114 Imo, Nigeria; osujyke@yahoo.com (C.M.O.); jude.iwounu@futo.edu.ng (J.O.I.); ookabuo@yahoo.com (N.O.K.); macmosesforall@yahoo.com (M.O.); ijeagunwah200@yahoo.com (I.M.A.); onyinye2121@gmail.com (O.P.M.); 2School of Food Science and Engineering, South China University of Technology, Guangzhou 510640, China; feyoulijun@scut.edu.cn (L.Y.); james.chacha@sua.ac.tz (J.S.C.); adedoyinodebade073128@gmail.com (A.O.A.); 3Food Technology Division, School of Industrial Technology, Universiti Sains Malaysia, Minden 11800, Penang, Malaysia; 4Department of Food Technology, Nutrition and Consumer Sciences, Sokoine University of Agriculture, 3006 Morogoro, Tanzania; 5Department of Food Technology, University of Ibadan, 200284 Ibadan, Nigeria; 6Institute for Biological Resources and Marine Biotechnologies—IRBIM, National Research Council (CNR), Via Vaccara, 61, 91026 Mazara del Vallo, Italy; giacomosardo88@gmail.com (G.S.); gioacchino.bono@cnr.it (G.B.); 7Department of Functional Food Products Development, Faculty of Biotechnology and Food Science, Wroclaw University of Environmental and Life Sciences, 51-630 Wroclaw, Poland; malgorzata.korzeniowska@upwr.edu.pl

**Keywords:** antioxidant capacity, biological/food systems, catalysed degradation, free radical, hydrogen peroxide, molecular modification, polysaccharide

## Abstract

Numerous reactive oxygen species (ROS) entities exist, and hydrogen peroxide (H_2_O_2_) is very key among them as it is well known to possess a stable but poor reactivity capable of generating free radicals. Considered among reactive atoms, molecules, and compounds with electron-rich sites, free radicals emerging from metabolic reactions during cellular respirations can induce oxidative stress and cause cellular structure damage, resulting in diverse life-threatening diseases when produced in excess. Therefore, an antioxidant is needed to curb the overproduction of free radicals especially in biological systems (in vivo and in vitro). Despite the inherent properties limiting its bioactivities, polysaccharides from natural sources increasingly gain research attention given their position as a functional ingredient. Improving the functionality and bioactivity of polysaccharides have been established through degradation of their molecular integrity. In this critical synopsis; we articulate the effects of H_2_O_2_ on the degradation of polysaccharides from natural sources. Specifically, the synopsis focused on free radical formation/production, polysaccharide degradation processes with H_2_O_2,_ the effects of polysaccharide degradation on the structural characteristics; physicochemical properties; and bioactivities; in addition to the antioxidant capability. The degradation mechanisms involving polysaccharide’s antioxidative property; with some examples and their respective sources are briefly summarised.

## 1. Introduction

Free radicals are a group of reactive atoms, molecules, and compounds with electron-rich sites produced as a result of metabolic reactions during cellular respiration [1,2]. In biological/food systems, free radical reactions are predominantly present and play a key role in biochemical pathways [1]. Collectively, free radicals are classified in two groups, namely reactive oxygen species (ROS) and reactive nitrogen species (RNS). The ROS involves oxygen-based highly reactive free radicals whereas the RNS involves nitrogen moieties associated with oxygen. In other words, RNS is a subset of ROS. Since oxygen radicals were discovered as harmful molecules [3], the causative agent of oxidative stress [4], and among essential free radicals generated during cellular metabolism, research interest on ROS has increased impressively. The ROS remains in the group of reactive oxygen-containing molecules possessing one or more unpaired electrons capable of existing independently [5]. The ROS comprises radical species and non-radical species. The former involves chemically reactive oxygen radicals such as hydroxyl, superoxide, peroxy, and alkoxy radicals, whereas the latter involves non-radical derivatives of oxygen such as hydrogen peroxide (H_2_O_2_,), hypochlorous acid (HOCl), singlet oxygen (O_2_), ozone (O_3_), and peroxynitrite (ONOOH) [6]. Moreover, non-radical species though not free radicals can still easily result in free radical reactions in living organisms, once conditions are favourable [7,8]. The ROS can be seen as a natural by-product of aerobic cellular metabolism with a double role, in physiological regulation and a key biological messenger in cell signalling cascades [9,10,11] such that when in low concentration, could cause oxidative stress, and when in excess or high concentration, could significantly damage the cell structures. Oxidative stress has always been hypothesized to be linked to excess production of ROS and/or impaired antioxidant defense capacity [12] or has also been perceived to occur due to an imbalance between the production of ROS and the available antioxidant or radical scavengers [13]. Over time, the accumulation of oxidative stress determined by the action of free radicals can cause lipid, protein, and DNA damage or alterations [7,14,15], as well as many life-threatening diseases including cancer, cardiovascular, Alzheimer disease, and related neurodegenerative diseases [11,16] diabetes mellitus, rheumatoid arthritis, cataracts, respiratory disease and aging process [7].

Numerous ROS entities exist, and H_2_O_2_ is very key among them as it is well known to possess stable [7,17] but poor reactivity [1] capable of generating free radicals. In addition, H_2_O_2_ is a metabolite of superoxide metabolic reactions [17] and a substrate for the production of hydroxyl radicals during cellular respiration in living organisms. Since its discovery, there has been a substantial increase in research interest regarding in vivo and in vitro formation mechanisms of H_2_O_2_—a ubiquitous endogenous molecule considered toxic to human tissues [18], and how it functions as a biomarker (target) in directing therapeutic delivery for oxidative stress-related diseases [19]. H_2_O_2_ is generated via respiratory chain cascade, but also a by-product of cellular metabolism [17]. Chemically, in contrast to superoxide anion (O_2_^•−^) and hydroxyl (OH•) radical, H_2_O_2_ is not very reactive [1] especially in the absence of transition metal ions and can act as a mild reducing and oxidizing agent. Though H_2_O_2_ does not readily oxidize proteins, lipids, and DNA [20], how it gets involved in such numerous physiological processes as cell signal transduction, cell differentiation (apoptosis), growth, and maintenance, as well as mediation of immune responses [17] demonstrates its dual role potential. Like other ROS molecules, and besides the dual functionality of H_2_O_2_, there are reported levels of H_2_O_2_ being cytotoxic to cells when the concentration is greater than 50 µM and limited cell cytotoxicity when concentration is at or below about 20–50 µM [20]. The effect of H_2_O_2_ in a living system is dependent on the type of cell, its concentration, its physiological state, and duration of exposure [17,20,21,22]. Interestingly, H_2_O_2_ can be part of redox signalling but can induce oxidative destruction of organs and tissues, associated with other inflammatory responses when overexpressed in cells [11,23]. Note worthily, H_2_O_2_ can produce very reactive OH• free radicals either in the presence of redox metal ions, such as copper or iron (Fenton reaction) or in the presence of oxygen radical (Haber-Weiss reaction) [24]. Hence, the need for antioxidants to scavenge free radicals from ROS which are causative agents of oxidative stress cannot be over-emphasized.

Antioxidants are substances able to inhibit oxidation caused by free radicals and unstable molecules produced in a living organism. They are applied to foods to inhibit free radical formation due to lipid oxidation and reduce the in vivo free radical concentration after (food) digestion [25]. The oxidative processes of antioxidants are underpinned by free radicals, being propagated by chain reactions in foods. For the most part, lipid oxidation (LOx) is sure to occur via mechanisms that involve complex processes such as autoxidation, enzymatic/thermal oxidation, and photo-oxidation. These processes are already well explained elsewhere [26,27]. For emphasis, antioxidants have two major groups that include: natural and synthetic types. Natural antioxidants have two major groups, namely, enzymatic and non-enzymatic antioxidants. Enzymatic antioxidants have primary (catalase, glutathione peroxidase (GPx), and superoxide dismutase) and secondary (glucose-6-phosphate dehydrogenase and glutathione reductase) enzymes, whereas non-enzymatic antioxidants have carotenoids, cofactors, flavonoids, minerals, phenolic acids, and vitamins. On the other hand, synthetic antioxidants can include butylated hydroxytoluene (BHT), butylated hydroxyanisole (BHA), *tert*-butylhydroquinone (TBHQ), and esters of gallic acid (e.g., propyl gallate) [27]. Nowadays, nutritional awareness has significantly influenced consumer demand and acceptance for safe, organic, and natural antioxidant sources. The preference for organic (food) products and/or additives such as antioxidants over synthetic ones by most consumers has increased interest in the use of antioxidants from natural sources. In addition, the components that make the antioxidant functions/mechanism work include its hydroperoxide destroyers, metal chelators, oxygen quenchers, radical scavengers, as well as (antioxidant) regenerators [27].

Polysaccharides are among the most abundant biomaterials which have attracted serious research interest in the field of pharmacology and biochemistry [25,28,29]. In recent years, polysaccharides obtained from natural (plants, animals, and microorganisms) sources have shown promising biological activity, such as antioxidation, antitumor, immunity, hypoglycaemic, and anti-aging properties, as well as cardiovascular, liver, and kidney protective effects [30,31,32,33,34], which is indicative of their high potential, especially in the field of medicine [25,35]. However, the high molecular viscosity/weight of many polysaccharides is considered to limit their pharmaceutical application. Modifying the chemical aspects of polysaccharides can therefore help achieve promising pharmacological/therapeutic targets [36,37]. The biological activities of polysaccharides may depend on several structural parameters, such as the branch structures, glycosidic linkages, and their types, molecular weight, and monosaccharide compositions. For instance, the β-(1→3) and β-(1→6) glycosidic linkages present within the repeating units of the polysaccharides have been deemed vital for anticancer activity [38]. Natural polysaccharides’ bioactivities and application are significantly influenced by the molecular weight of the native polysaccharide. For instance, a polysaccharide with larger molecular weight, on the one hand, may not easily penetrate cell membrane barriers in organisms and this could result in impaired biological effects [34,39,40]. On the other hand, Wu et al. [41] have reported higher biological activity in low molecular weight natural polysaccharides. Thus, finding efficient methods for the degradation of natural polysaccharides is crucial, if the biological activity of polysaccharides were to be facilitated. The physical, chemical, enzymatic, and/or a combination of these modification methods could substantially enhance both characteristics and utilities of (natural) polysaccharides.

## 2. Justification of Synopsis and Objective Statement

Ranked third among the macromolecules after nucleic acids and proteins that carry essential biological information, polysaccharides and their associated (protein) complexes from natural sources (such as bacteria, fungi, algae, and plants) continue to display effectiveness as non-toxic substances, demonstrated by diverse biological functions [42]. It is important to reiterate that among the active polysaccharides from natural sources, those that are sulfated are considered acceptable and safe [43]. Various polysaccharide extraction methods are possible due to novel techniques like enzyme-assisted extraction (EAE), microwave-assisted extraction (MAE), as well as ultrasonic-assisted extraction (UAE). Additionally, polysaccharide extraction has largely focused on both effects as well as the yield of a single variable, not forgetting its activity (that is, polysaccharides) [44]. However, considering the existence of ROS such as H_2_O_2_, the presence of free radicals that induce oxidative stress, and the need for an antioxidant to curb its (free radicals) overproduction within biological systems (in vivo and in vitro), it is important to understand how the aforementioned (physical, chemical, and/or enzymatic modification) methods can extend the characteristics and utilities of (natural) polysaccharides.

The chemical method of polysaccharide modification (degradation) via oxidation has been of increasing research interest, given the contributions to this body of knowledge specific to the degradation of polysaccharides via oxidation method largely by the H_2_O_2_ method. Although studies focused on the degradation of polysaccharides by free radicals increasingly grows, there is a paucity of literature that has synthesized the polysaccharide degradation by free radicals from H_2_O_2_, to particularly demonstrate its potential use in the agro-food industry, together with it being a viable candidate for in vivo biological activity especially when incorporated within the food matrix/system. On this premise, this current critical synopsis articulates the H_2_O_2_ effects on the degradation of polysaccharides from natural sources. Specifically, the structure of this synopsis will give emphasis to free radical formation/production, the effects of polysaccharide degradation with H_2_O_2_ on the structural characteristics, physicochemical properties, and bioactivities, in addition to the antioxidant capability, and reaction mechanism.

## 3. Survey Methodology

To actualise this critical synopsis, the first step taken was to formulate the research questions, ensuring the contribution of this review is relevant to this field, and at the same time, considering the intended/target audience. In the next step, after having agreed with the research questions, we developed a search strategy. This allowed for the establishment of both inclusion and exclusion criteria, which determined the appropriate research articles, and considered relevant specific to degradation of polysaccharides by free radicals from H_2_O_2_. Based on this chosen topic, the search terms such as polysaccharides, natural antioxidants, free radicals, ROS, polysaccharide degradation, H_2_O_2_ (formation/sources and reaction mechanism), and natural polysaccharides were identified and selected.

Published papers and their references were collated from databases like EMBASE, Google Scholar, Web of Science, Interscience Online Library, MEDLINE, ScienceDirect, and PubMed. By way of inclusion and exclusion criteria, the research articles not addressing the research question(s) were expunged, considering the year of publication and the language of the article (publications in languages other than English were excluded). Importantly, the data from the literature of selected studies were analysed and relevant information incorporated helped to expatiate the discourse of this conducted critical synopsis.

## 4. Discussion of Findings

### 4.1. H_2_O_2_ Formation and Factors Influencing its Production

The H_2_O_2_ remains a key affiliate of the group of ROS next to superoxide anion (O_2_^•−^) and OH• radicals, which is produced through the respiratory chain cascade and a by-product of cellular metabolism [17,45,46]. As a member of ROS, its reactivity targets proteins, lipids, and DNA alteration or damage, which brings about oxidative stress [47,48,49]. H_2_O_2_, as a crucial second messenger in normal cellular signalling [11], has had its overproduction associated with various life-threatening diseases [2,6,11,20] since it is considered cytotoxic [1,20]. Besides being the simplest form of peroxide [5] having two oxygen atoms covalently bonded (single bond), it is endogenously produced in diverse cellular aspects of tissues or organs (peroxisomes, endoplasmic reticulum, and mitochondria) at high levels of oxygen utilisation [7].

However, many studies have reported that H_2_O_2_ is a metabolite of O_2_^•−^ produced in the mitochondria during cellular respiration propelled by electron transport chain (ETC) [6,7,17], and a substrate for the production of OH• radical during cellular metabolism [7]. The O_2_^•−^ (half-life of 10^−6^s), OH• radical (half-life of 10^−10^s), and H_2_O_2_ (stable half-life) are key members of ROS [7]. H_2_O_2_ as a non-radical is fairly reactive (in the absence of transition metal ions) probably due to its stable half-life, but its reactivity is enhanced when it reacts with transition metal ion to yield OH• radical (Fenton reaction), as well as when it reacts with O_2_^•−^ to yield OH• radical (Haber-Weiss reaction) [24].
(1)$$\begin{array}{c} \mathrm{Fenton}\;\mathrm{reaction}:\\ {\mathrm{Fe}}^{3+}+{O_{2}}^{{•}^{-}}\;\to \;{\mathrm{Fe}}^{2+}+O_{2}\;(\mathrm{auto-oxidation})\\ {\mathrm{Fe}}^{2+}+H_{2}O_{2}\;\to \;{\mathrm{Fe}}^{3+}+{\mathrm{OH}}^{-}+\mathrm{OH}• \end{array}$$
(2)$$\begin{array}{c} \mathrm{Haber-Weisss}\;\mathrm{reaction}:\\ H_{2}O_{2}+{O_{2}}^{{•}^{-}}\;\to \;O_{2}+{\mathrm{OH}}^{-}+\mathrm{OH}• \end{array}$$


In the first instance (Equation (1)), the Fenton reaction begins with the production of superoxide radical via cellular metabolism. During cellular respiration, some electrons that escaped the electron transport chain can interact with oxygen (O_2_) to yield (unstable) O_2_^•−^, a member of ROS. As a non-enzymatic product, the generation of superoxide anion (O_2_^•−^) increases with an increase in metabolic rate, which causes cellular components destruction. Consequently, low amounts of superoxide anion (O_2_^•−^) are maintained in the cells with the aid of a metalloprotein enzyme [2] called superoxide dismutase (SOD), which catabolizes superoxide anion (O_2_^•−^) to H_2_O_2_ and subsequently catabolized (detoxified) by the enzyme catalase (metalloenzyme) and glutathione peroxidase (GPx) [7] to water and oxygen.

As shown in Figure 1, there is the electron transport chain, which involves a series of electron transporters embedded within the inner mitochondrial membrane that shuttles the cascade of electrons from the NADH and FADH_2_ to molecular oxygen. It is at this process that protons are pumped from the mitochondrial matrix to the intermembrane space, and oxygen is reduced to form water. Specifically, O_2_^•−^ levels may rise, at times, after the metabolism of alcohol by actions of alcohol dehydrogenase (ADH) enzyme, and NAD^+^ generates a lot of nicotinamide adenine dinucleotide (NADH) [50]. As a result, more H_2_O_2_ is produced which may not be either detoxified and or hydrolysed, by the limited amount of enzyme (catalase and glutathione peroxidase). Moreover, the H_2_O_2_ would react with the transition metal ions, such as iron (Fe^2+^) and copper (Cu^+^), respectively found in the cells as ferritin and ceruloplasmin [51], and in that way, able to generate OH• radical, which remains a highly unstable molecule [52,53].

On the other hand, the Haber-Weiss reaction (Equation (2)) mechanism, which generates OH• radical from H_2_O_2_ and O_2_^•−^ catalysed by iron as described by Fritz Haber and his student Joseph Joshua Weiss [5] is considered as among the key foundation of free radical biochemistry [53]. Figure 2 shows the role of H_2_O_2_ in cell dysfunction and transformation. Notably, both the Fenton and the Haber-Weiss reactions have proven the role of H_2_O_2_ as an intermediate and substrate respectively, for the production of OH• radicals responsible for oxidative stress and cellular damages [5,53]. The OH• radical, as a neutral form of hydroxyl ion (OH^−^), remains the most reactive member of the ROS because of its short-lived half-life, estimated to be about 10^−10^ s. The destructive action of this unstable molecule (OH•) is implicated in series of cell dysfunction, transformation, and even death [5,54,55].

#### 4.1.1. Formation of H_2_O_2_

The formation of H_2_O_2_, a member of ROS, is the outcome of an unavoidable aerobic metabolism or reaction [56]. Many studies have understood H_2_O_2_ as poorly reactive yet somewhat stable. However, the destructive activity and cytotoxic effects of H_2_O_2_ have been owed to the formation of OH• radical, when it reacts with either superoxide anion(s) or transition metals. The formation/production of H_2_O_2_ in vitro and in vivo [57] can, respectively, take place in certain beverages and humans.

(a)In vitro production of H_2_O_2_

The production of H_2_O_2_ by various food systems has been reported previously. For instance, studies have shown H_2_O_2_ production from thiol-rich proteins during barley (malt) beer production [58] and in the association of L-cysteine with Cu(II)-catalysts during malting and beer brewing [59]. Also, H_2_O_2_ can be generated in vitro in certain polyphenol-rich beverages such as green tea, black tea [60,61,62], red wine, and coffee [60,63]. Polyphenols are a large group of compounds naturally found in fruits, vegetables, and certain beverages, possessing antioxidant properties. The protective effect of polyphenols which is attributed to its antioxidative property is believed to be due to its ability to act as a free radical scavenger, quenching the activities of ROS such as O_2_^•−^ radical, OH• radical, and H_2_O_2_ [62,64]. In contrast, the production of H_2_O_2_ by polyphenols understood from the (polyphenol) pro-oxidant action via autoxidation [60,65] has been associated with induced mutagenesis and promotion of cancer [62], as well as to promote oxidative damage to DNA, lipids, and proteins in the presence of transition metal ions under certain conditions in vitro [66].

The phenolic compounds that produce H_2_O_2_ through the autoxidation process include tea catechins (epicatechin [EC], epicatechin gallate [ECg], epigallocatechin [EGC], and epigallocatechin gallate [EGCg]) [61] (+)-catechin, and gallic acid [67]. Further, Grzesik et al. [63] reported that (black and green) teas and coffee but not cocoa were shown to produce considerable concentrations of H_2_O_2_. In such beverages (green tea, black tea, and coffee), the H_2_O_2_ generating property is poised to coincide with the content of phenolic compounds. This suggested that polyphenols are liable for H_2_O_2_ production in beverages. Results from previous researches indicated that the formation of H_2_O_2_ from the aforementioned phenolic compounds in an aqueous solution would depend on the content and composition of polyphenols, structure of the phenolic compound, oxygen, metal ions, temperature, pH, and incubation time [66]. Therefore, the combined effect of these conditions would result in the formation as well as diverse concentration(s) of in vitro H_2_O_2_ production.

(b)In vivo production of H_2_O_2_

The H_2_O_2_ can be naturally produced in vivo for example, in humans, as a by-product of oxidative metabolism [5] by the dismutation of superoxide radical, both non-enzymatically and catalysed by superoxide dismutase (SOD) enzymes [68]. Besides the superoxide dismutase (SOD) enzyme, there are other oxidase enzymes such as glycollate and monoamine oxidases, directly linked to the production of H_2_O_2_ [68,69]. Additionally, the glutathione peroxidase and thioredoxin-linked peroxidase are enzymes contained in the mitochondria. However, the effectiveness of H_2_O_2_ removal by the enzymes remains unclear, given the rate at which it is generated in the mitochondria [70,71]. Although certain tissues may be exposed to higher H_2_O_2_ concentrations with the mitochondria being at the helm of activity in human cells, most cells may also be exposed to certain H_2_O_2_ levels.

The formation of H_2_O_2_ in the oral cavity, oesophagus, stomach, respiratory system, kidney, urinary tract and bladder, vascular endothelial and circulating blood vessel as well as ocular tissues are well documented. Lambert et al. [72] reported the in vivo generation of H_2_O_2_ in micromolar concentrations produced when green tea solution is held in the mouth. Halliwell et al. [68] reported that beverages like green tea, black tea, and instant coffee contain H_2_O_2_ at concentrations above 100 µM, probably because of the joint action of autoxidation of antioxidants in the beverages and oral bacteria. Production of H_2_O_2_ by oral bacteria has been previously reported, although its impact on oral tissues remains unknown. Moreover, H_2_O_2_ can be present in exhaled air of healthy humans although its concentration is higher in those with inflammatory lung diseases and in cigarette smokers [68]. This H_2_O_2_ in exhaled air can possibly be produced by oral bacteria since cells lining the respiratory system are exposed to about 21% oxygen. Besides H_2_O_2_ detected in freshly voided human urine [68], it could be at a higher concentration in the urine of coffee drinkers [63]. The presence of H_2_O_2_ in human urine may be due to the dismutation of superoxide radicals which may come from the diet via autoxidation. Adahi et al. [73] detected some traces of superoxide dismutase (SOD) in the urine. Additionally, the exhalation of H_2_O_2_ and its removal through urine could be part of the human excretory mechanism.

The in vivo generation of H_2_O_2_ in blood plasma by the action of xanthine oxidase enzyme is degraded in part by traces of catalase. Like in blood plasma, ocular fluids have been reported to generate H_2_O_2_ constantly which is usually rapidly removed [74,75]. Though the origin of ocular H_2_O_2_ is unknown, oxidation of ascorbate may likely be a contributory factor [76]. Importantly, it is expected that H_2_O_2_ would accumulate as a result of the joint action of in vivo dismutation of superoxide radical and in vitro autoxidation of phenolic compounds (antioxidants). However, there is no evidence that these beverages (teas and coffee) promote or induce mutagenesis and carcinogenesis in vivo. Possibly, any threat the in vitro generated H_2_O_2_ from polyphenols is likely to pose could be inhibited by phenolic compounds and other constituents of the beverages [60], or even catalase and glutathione peroxidases in the body, and decomposed by the reaction with biomolecules [62]. In a research on dietary antioxidants as a source of H_2_O_2_, Grzesik et al. [63] reported that 27 out of 52 antioxidants generated H_2_O_2_, suggesting the availability of other constituents capable of exerting antioxidant property in the beverage.

#### 4.1.2. Factors That Influence the Production of H_2_O_2_

As was earlier mentioned, the in vivo and in vitro formation of H_2_O_2_ is dependent on some factors that could inhibit or enhance its production, and they include the composition and structure of phenolic compounds, oxygen, metal ions, pH, enzyme, and incubation condition (temperature and time).

(a)Composition and structure of phenolic compounds

According to Akagawa et al. [62], only *o*- and *p*-phenolic compounds undergo autoxidation. Additionally, EGC, EGCg, and compounds with gallic acid moieties have been reported to produce H_2_O_2_ during oxidation in aqueous media [61,77,78,79] while Akagawa et al. [62] reported that the formation of H_2_O_2_ by tea catechins followed the order: epigallocatechin gallate > epicatechin gallate ≡ epigallocatechin > epicatechin > catechin.

Phenolic compounds with OH-group, substituted at the 2 and 4 positions of phenol, would generate H_2_O_2_ [80]. Such production would significantly increase by the insertion of a third OH-group. This, therefore, corroborates the relationship between the H_2_O_2_ generating property of phenolic compounds and their structures [61] (Figure 3). Akagawa et al. [62] reiterated this, in that the insertion of a third adjacent hydroxyl group, that is, the presence of gallate substituent (pyrogallol moiety) in the catechins, would increase the formation of H_2_O_2_. In practice, a significant amount of H_2_O_2_ can be produced by gallic acid. The H_2_O_2_ generating property in these beverages was found to coincide with the content of phenolic compounds, indicating that polyphenols are responsible for the production of H_2_O_2_ in beverages.

(b)Oxygen and metal ions

Oxygen is essential for the viability and maintenance of cell metabolism but has been found to play a key role as a potential promoter of in vitro and in vivo free radical reactions [81,82]. Electronically it is an highly excited molecule that exists in two excited states upon activation. In the mitochondria, the formation of H_2_O_2_ is initiated by oxygen during the mitochondrial respiratory chain, where about 85% of oxygen is metabolized. The transition metal ions mainly considered by researchers are iron, manganese, copper, and sometimes zinc because these species are major players in free radical or electron transfer reactions [1]. The joint action of transition metal ions and oxygen in the production of H_2_O_2_ has been reported in many studies. According to Mochizuki et al. [83], tea polyphenols are easily oxidized to the corresponding quinones by oxygen (O_2_) in the presence of transition metal ions, incidentally resulting in the formation of O_2_^•−^ and H_2_O_2_ through autoxidation (Figure 4). In addition, higher H_2_O_2_ formation has been detected in tea infused in tap water containing transition metal ions than a tea infused with deionised water [63], thus suggesting that the autoxidation of polyphenols would produce H_2_O_2_, catalysed by transition metal ions. The autoxidation of polyphenols by oxygen to yield semiquinone radical and O_2_^•−^ in the absence of metal ions was reported by Akagwa et al. [62] though with a slow reaction rate due to the low redox potential of the O_2_/O_2_^-^. On the other hand, metal ion (Cu^2+^) catalysed autoxidation of polyphenols to yield semiquinone radical and Cu^+^ have also been reported. Subsequently, O_2_^•−^ and quinone is produced when the generated semiquinone radical is rapidly oxidized by oxygen [83]. Hence, the generated O_2_^•−^ helps to make the H_2_O_2_ spontaneously via dismutation [62], and also to H_2_O_2_ and semiquinone radical via oxidation of polyphenols.

(c)pH

Multiple studies have reported the correlation between pH and H_2_O_2_ formation from tea polyphenols. Mochizuki et al. [83] reported that the autoxidation of polyphenols depends on the pH of the medium (tea) and is accelerated by alkaline pH. Similarly, Grzesik et al. [63] reported that the rate of autoxidation of compounds increased with pH, which corroborated with findings of Rinaldi et al. [84] and Akagawa et al. [62] wherein the strong electron-donating ability of acid dissociated the phenolic compound, as polyphenols underwent autoxidation at a higher rate with increased pH. Nakayama et al. [61] detected H_2_O_2_ in polyphenol-rich beverages (tea and coffee) when the pH value is above 5.8 and its concentration increased with pH up to 8.0. A maximum formation of H_2_O_2_ at pH 7.0–9.0 for tea polyphenols has also been reported [62]. However, the addition of milk and lemon respectively decreased and prevented H_2_O_2_ formation in tea [60,63]. This was shown by the lower pH ranges in milk (pH 6.5–6.7) and lemon (pH 2–3), which suggested H_2_O_2_ formation as pH-dependent, where a decrease in pH reduces or prevents its (H_2_O_2_) formation and vice versa.

(d)Incubation condition

The incubation condition of tea polyphenols plays a significant role in H_2_O_2_ formation. The action of temperature-time incubation conditions cannot be over emphasised. Studies have shown that an increase in incubation temperature and time increases the rate of H_2_O_2_ formation. In the same vein, Nakayama et al. [61] reported an increase in H_2_O_2_ formation when the incubation temperature of tea was increased from 37 °C to 77 °C. However, Nakayama et al. [66] reported that the concentration of H_2_O_2_ in tea increased concavely with temperature when increased from 32 °C to 82 °C. Also, the amount of H_2_O_2_ formation in tea has been reported to increase almost linearly with incubation time [66], suggesting that autoxidation of polyphenols is also time-dependent.

(e)Enzymes

Enzymes have been found to participate in the production of H_2_O_2_. Some H_2_O_2_-producing enzymes include but are not limited to superoxide dismutase (SOD), and other oxidase enzymes such as glycollate and monoamine oxidase [68,69], glucose oxidase, galactose oxidase, and cholesterol oxidase [85]. Superoxide dismutase catabolizes O_2_^•−^ to form H_2_O_2_ [7], while glucose oxidase catalyses the oxidation of β-D-glucose by oxygen to form H_2_O_2_ and gluconolactone (Figure 5) [85]. Also, cholesterol oxidase catalyses the oxidation of cholesterol by oxygen to yield H_2_O_2_ (Figure 6) while galactose oxidase catalyses the oxidation of primary alcohol and polysaccharides (for example, D-galactose) and primary alcohols by oxygen, resulting to corresponding aldehydes and H_2_O_2_ as shown in the equation below:RCH_2_OH + O_2_ → RCHO + H_2_O_2_(3)

Notably, some key situations associated with the formation of H_2_O_2_ which can combine together via series of chain reactions under both in vitro and in vivo conditions are briefly presented in Table 1. In the mitochondria (in vivo), oxygen combines with an escaped electron in the electron transport chain to form O_2_^•−^, which is subsequently oxidised to H_2_O_2_ by superoxide dismutase (SOD). This implies that the formation of H_2_O_2_ in the mitochondria is dependent on the O_2_^•−^, which may increase in concentration with an increasing amount of NADH. On the other hand, in beverages (in vitro) such as tea and coffee, autoxidation of polyphenols by oxygen, catalysed by transition metal ions to yield corresponding quinones (depending on the composition and structure of phenolic compound) subsequently generates H_2_O_2_, whose concentration is dependent on the beverage incubation condition (temperature and time).

### 4.2. Polysaccharide: The Degradation Process and Effects with H_2_O_2_

Polysaccharides are the most common biopolymers and copious organic material in existence [25,92,93]. For instance, the carbohydrate molecules of carbon and oxygen atoms differ in size and range from one to thousands of units. Simple sugars (monosaccharides) are the smallest unit of a polysaccharide molecule. Examples include glucose, fructose, mannose, and galactose. Polysaccharides are broadly classified as homo-polysaccharides, which can include, one type of monosaccharide units like glycogen, cellulose, and starch, and the hetero-polysaccharides, which can include two or more types of monosaccharide units like hyaluronic acid, heparin, etc. Furthermore, cellulose is a structural polysaccharide found in plants that forms the main structural component of plant cell wall [25] whereas glycogen and starch are storage polysaccharides found in animals (muscles and liver), and plants respectively. Besides the well-known animal/plant polysaccharides, there are other less popular polysaccharides that arise from sources like bacteria, fungi, and algae [25].

Ever since polysaccharides were discovered to possess some biological activity, there has been increasing research interest on this specific aspect in the field of biochemistry, life science, pharmacology, and medicine, together with its possible applications [25,94]. Garcia-Gonzalez et al. [95] highlighted the use of polysaccharides as key functional ingredients in the production of bio-based materials in food, cosmetics, medical devices, and pharmaceutics. However, polysaccharides obtained from plants, animals, and microorganisms have been shown to possess diverse properties not limited to antioxidative, anti-inflammatory, antimutagenic, antitumor, health-promoting, and therapeutic properties [40,94,96,97,98]. With the many bioactivities/properties exerted by polysaccharides, our focus in this current review was on the resurgent antioxidative effect of polysaccharides, which can be explored as a potential novel functional food additive in food formulation.

#### 4.2.1. Polysaccharide Degradation

There is evidence that, despite exerting antioxidative properties, not all polysaccharides have bioactivity, which is the situation, for example, of untreated pachymaran [99]. Additionally, some polysaccharides might not exhibit any bioactivities given their structural and physicochemical properties [94]. Many studies have shown that the functional role of polysaccharides is dependent on their degree of polymerization and molecular weight [46]. This was reiterated by Li and Shah [100] wherein the bioactivities of most polysaccharides were influenced by molecular weight and structural characteristics such as monosaccharide compositions, linkage patterns, anomeric carbon configuration, branching features, degree of polymerization, and sequence of sugar units. Despite these factors, several studies have shown that polysaccharides’ bioactivities are significantly influenced by the molecular weight of the native polysaccharide because polysaccharides with too high (large) molecular weight may not easily penetrate cell membrane barriers in organisms due to their high viscosity and low solubility, and will therefore result to impaired biological effect [34,39,40,94,101]. Hence, the physicochemical properties and bioactivities of polysaccharides are significantly influenced by their structure; so the structure of polysaccharides must be modified via degradation to the desired molecular weights that will enhance the bioactivities of polysaccharides.

#### 4.2.2. Methods of Polysaccharide Degradation

The growing realisation of the potential novel products with diverse applications which may be obtained from degradation (depolymerisation) of the structural integrity of polysaccharides gave rise to the development of different degradation methods [102]. Generally, the degradation of polysaccharides can be achieved via physical, chemical, and enzymatic methods [40,102,103], as well as biological means [94], to obtain a diversity of structural derivatives. Additionally, some other methods of polysaccharide degradation such as thermal, ultrasonic, mechanical, lyophilisation-mediated, metal-catalysed, free radical-catalysed [102], and irradiation [104] degradation have been previously described. Besides the advantages of primary degradation methods such as being simple to operate or easy to handle (chemical method), environmentally friendly (physical method), low cost, good safety, and mild reaction condition (biological method), each method also has its limitations. Multiple studies have reported environmental pollution and serious safety concerns (chemical method), utilization of specialized equipment which limits their industrial application (physical method), and difficulty in the screening of enzymes and microbes (biological method), as some of the challenges attributed to these treatment methods for polysaccharide degradation [105,106,107,108]. With the aforementioned methods of polysaccharide degradation, the chemical degradation through the oxidation method (either singly or combined) continues to attract researchers’ interest. For example, Zhao. [34] showed that the free radical-catalysed H_2_O_2_ method is frequently employed when they investigated polysaccharides from *Tremella fuciformis*. Furthermore, the fragmentation (degradation) of polysaccharides by ROS has been increasingly evidenced, as the bioactivities and overall functionality (of polysaccharides) gets altered [46]. Noteworthily, treating polysaccharides with H_2_O_2_ (only) does not generally degrade its high molecular weight, instead, the activation of H_2_O_2_ is necessary to produce highly reactive OH• radicals that can enhance its degradation property [106].

H_2_O_2_, as a member of ROS, during polysaccharide degradation operates under mild reaction conditions, with high efficiency, despite its inability to compromise the integrity of the main chain (polysaccharide) structure [40,98,109]. However, the various degradation methods employed are mediated by different reaction mechanisms, to yield potentially distinct products. Recently, desirable modified polysaccharides have been obtained with higher bioactivities by combining different degradation methods [94]. According to Wu et al. [104], the synergistic effect of combining different polysaccharide degradation methods significantly affects the depolymerisation kinetics of the process. Based on this, several combined polysaccharide degradation methods developed include H_2_O_2_-Vitamin C, H_2_O_2_-UV, H_2_O_2_-NaClO_2_, H_2_O_2_-HA_c_, Fe^2+^-H_2_O_2,_ Fe^2+^-H_2_O_2_-vitamin C, Solution plasma process (SPP) irradiation, and SPP irradiation–H_2_O_2,_ Ultrasonic irradiation-H_2_O_2_ [34,36,40,104,110,111,112], as well as a compound treatment of H_2_O_2_-vitamin C-microwave treatment [107]. In other words, H_2_O_2_ serves as an important source of reactive oxygen species (ROS) for these combined treatments, to assure adequate production of OH• radicals for significant degradation of polysaccharides.

#### 4.2.3. Effect of Polysaccharide Degradation with H_2_O_2_

H_2_O_2_ continues to be a promising candidate for polysaccharide degradation. Polysaccharide degradation (depolymerisation) with H_2_O_2_ or in combination with another technique, by changing its functional role, would initiate its (polysaccharide) modification. Largely, the H_2_O_2_ is used in polysaccharides degradation given that it is easily available, easy to handle, and environmentally friendly since it decomposes to water and oxygen [106,113] as well as its ability to form reactive OH• radical through Fenton reaction [104]. In free radical-catalysed depolymerisation, the hydroxyl radicals via bond (glycosidic linkage) cleavage on the polysaccharides chain [102], would bring about molecular modification. Consequently, the modification of polysaccharides would lead to changes in physicochemical (relative molecular mass, viscosity, and solubility) and structural properties, as well as improved bioactivities. Li et al. [94] obtained enhanced biological properties in degraded/modified polysaccharides. Thus, both physicochemical and structural properties will be tersely discussed below:(a)Physicochemical Properties

Polysaccharide degradation can cause numerous changes in the physicochemical properties, from a decrease in its crystallinity as a result of structural rearrangements [114] to the improved tensile strength of its chitosan polysaccharide [94,115]. To understand the effect of polysaccharide degradation on the physical and chemical property, we attempt to examine aspects like its molecular weight, intrinsic viscosity, and water solubility, which are discussed briefly below.

(i)Molecular weight

The molecular weight of polysaccharides is the basis for their bioactivities. Polysaccharides with high molecular weight are associated with low bioactivity, due to the compactness of their structures, low surface area, and stronger intramolecular hydrogen bonds [30]. Interestingly, the intrinsic viscosity and water solubility of polysaccharides depends significantly on their molecular weight, which could indicate a clear correlation between these properties. High molecular weight polysaccharides could lead to increased viscosity and decreased water solubility. On this premise, the degradation of high molecular weight polysaccharides could result in a modified (polysaccharide) type, able to easily permeate the cell membrane barriers of organisms, which could, in turn, exert improved bioactivity. Suggestively, the bioactivity of polysaccharides significantly would depend on their molecular weight.

The molecular weight of degraded or modified polysaccharides would decrease significantly, compared to the native (unmodified) polysaccharide. For example, Zhang et al. [34] reported a successful degradation of high molecular weight polysaccharides from 599,580 to 26,895 Da. In addition, the rate of polysaccharide degradation into lower molecular weight (degraded) polysaccharide significantly depends on the concentration of degradation reagent (H_2_O_2_). A high molecular weight *Astragalus* polysaccharide (11,033 Da) treated with different concentrations of H_2_O_2_ (4%, 6%, and 14%) was significantly modified (degraded) to lower molecular weight polysaccharides of 8376, 4716, and 2600 Da, respectively [30]. Therefore, whilst degradation of polysaccharides enhances their bioactivities, the rate of degradation into lower molecular weights is determined by the concentration of H_2_O_2_ which was utilized during the degradation. Notwithstanding that low molecular weight polysaccharides can exert added bioactivities, degraded polysaccharides with very low molecular weight, on the other hand, might bring about a reduction or loss in polysaccharide bioactivity as the inactive polymeric structure develops [101]. That is to say, the destruction of the polysaccharide ring structure by the H_2_O_2_ degradation system in very low molecular weight polysaccharide also breaks its intramolecular hydrogen bond, making the carbonyl group be in an open chain, thereby resulting in reduced bioactivity.

(ii)Intrinsic viscosity

Multiple studies have shown the correlation between molecular weight and intrinsic viscosity. Due to the high molecular weight of polysaccharides, the high intrinsic viscosity of polysaccharides limits their absorption and utilization in vivo [40,94]. Notably, the intrinsic viscosity of degraded or modified polysaccharides decreases significantly with a decrease in molecular weight compared to the native polysaccharide. Xu et al. [40] observed a decrease in intrinsic viscosity of degraded polysaccharides from 2.26 to 1.43 mPa.s. Similarly, Wu et al. [104] observed a reduction in intrinsic viscosity of chitosan polysaccharides by 82.19% and 70.04% when the chitosan polysaccharide was treated with SPP irradiation-H_2_O_2_ and SPP irradiation degradation method, respectively. This implies that the rate of reduction of intrinsic viscosity of polysaccharides is higher when two or more polysaccharide degradation methods are combined. Zhang et al. [34] observed a successful degradation with a 77.07% decrease in the intrinsic viscosity of polysaccharides, from 285.24 to 65.39 mL/g using Fe^2+^-H_2_O_2_-Vc degradation method. Similarly, Li et al. [94] observed a decrease in intrinsic viscosity of modified polysaccharide solution from 34.67 to 22. 92 mL/g which was believed to be attributed to a reduction of the hydrogen bonds. It is evident that the molecular weight of polysaccharides is concordant to their viscosity. This was consistent with Zhang et al. [36] who reported that the decrease in viscosity of polysaccharides is attributed to the reduced interaction between macromolecules, due to a slight decrease in their molecular weight.

(iii)Water solubility

Most bioactive polysaccharides have not found useful applications because of their poor water solubility, a result of their high molecular weight [40,94]. On the other hand, many studies have recorded a significant increase in water solubility of polysaccharides after degradation or depolymerisation,. Xu et al. [40] found that the solubility of degraded polysaccharides increased from 18.24 to 24.40 mg/mL with a decrease in molecular weight. Free radical-catalysed degradation offered by H_2_O_2_ has been found to increase polysaccharide solubility because of the inauguration or incorporation of carboxyl groups, ionic groups, or other hydrophilic groups into the polymer structure [94]. In addition to increased water solubility, degraded polysaccharides had a greater surface area which translates to higher reaction rates [34,98]. Indeed, the increase in water solubility of degraded polysaccharides would bring about a significant transformation of polysaccharides especially to their application, to exert improved bioactivities.

(b)Structural Properties

The structure of polysaccharides influences their biological activity [30]. Polysaccharide degradation can produce different types of structures with varying bioactivities which lays a solid foundation for the functionality of the degraded or modified polysaccharide [94]. Free radical-catalysed degradation of polysaccharides offered by H_2_O_2_ reduces the association of amylose molecules by introducing small amounts of carbonyl and carboxylate groups [116] into the polymer structure which can improve the hydrophilic properties of the modified or degraded polysaccharides [94,117]. However, the surface area of the fragmented structure of degraded or modified polysaccharides could decrease significantly compared to its native polysaccharides, specifically due to the destruction of the aggregates or original interconnection in the polysaccharides by free radical-catalysed treatment offered by H_2_O_2_ [40,44]. Generally, by using only H_2_O_2_ or in combination with other degradation techniques, the main chain structure in the degraded or modified polysaccharides may not alter, but the side chain structure can change [30]. The degradation of chitosan structure, according to Tian et al. [118], only alters the side groups, but no significant change took place within the main chain structure. Some enhancement can take place at the side groups on the main chain polysaccharide structure, such as hydroxyl (–OH) and carboxyl (-COOH) groups, especially after the polysaccharide degradation. This can happen owing to the exposure of its effective active sites (groups), which thereby increases its content. Moreover, Wang et al. [30] detected a slight increase in degraded polysaccharide after the polysaccharide chain was broken by the OH• radical produced by the H_2_O_2_ degradation system, thus exposing the –COOH groups, whereas Zhao et al. [119] detected high levels of –COOH group in degraded polysaccharides. Similarly, Chang et al. [120] showed that a polysaccharide with a molecular weight of 129 KDa, degraded to lower molecular weights of 60 and 52 KDa, recorded an increase in their uronic acid content from 70.8% before degradation to 79.5% and 86.2% in the degraded polysaccharide, respectively. This was also consistent with the findings of Xu et al. [40] and Wu et al. [104]. Additionally, the helical structure of the main chain and hydrophilic group (hydroxyl group) influences the polysaccharides’ bioactivities. Also, polysaccharides with compact structures tend to possess stronger intramolecular hydrogen bonds, leading to less accessibility or exposure of its effective active sites (groups). To shed more understanding on this, we tersely discussed polysaccharides’ bioactivities and their workings.

(c)Bioactivities

There is increasing evidence that the functionality of polysaccharides such as their bioactivities can be enhanced only when their solubility in water increases, and this can be achieved via degradation or modification of polysaccharides [121]. In particular, the increase in hydrophilic properties of degraded polysaccharides, which has been attributed to their decreased molecular weight, lowered intrinsic viscosity, increased surface area, and structural changes (incorporation of carboxyl groups into the polymer structure), makes these degraded polysaccharides soluble in water, and as a result, exerts better bioactivities. Despite the many bioactivities, which include but are not limited to, antioxidant activity, antitumor activity, anti-inflammatory activity, anti-HIV activity, anticoagulant activity, antibacterial activity, antiviral activity, and immunomodulatory activity [94,121], our focus in this current literature synthesis is on the antioxidant activity of polysaccharides. Some researchers, however, have demonstrated that improved antioxidant activity of polysaccharides would associate with its degree of water-solubility [34,37]. To understand the workings of bioactivities, we shed some light on polysaccharides’ antioxidant activity in terms of their radical scavenging capacity, reducing power, and metal ion chelating ability.

(i)Antioxidant activity

Antioxidants are substances or compounds that inhibit oxidation caused by free radicals and unstable molecules produced in food or living organism. Antioxidants are specific in their action by performing their primary function of breaking chain reactions through free radical scavenging and also by performing their secondary function (preventive role) through chelation of metal ions that catalyse a free radical reaction. The antioxidant activity of substances or compounds is displayed through series of reaction mechanisms which include transition metals chelating ability, hydrogen electron transfer, and single electron transfer [81]. However, much research has shown that various polysaccharides obtained from plants, animals, fungi, bacteria, and algae possess antioxidant properties [94], especially when such a polysaccharide is the degraded or modified type. Specifically, polysaccharides are compounds with large molecular weights and as a result would have lesser non-reducing end contents, hence, weaker antioxidant activity [30]. Based on this fact, the degradation of polysaccharides can effectively better their antioxidant activity. Furthermore, series of studies on free radical-catalysed polysaccharide degradation have shown that the antioxidant activity of degraded polysaccharides was ascertained by determining their radical (OH•, O_2_^•−^, DPPH^•^, and ABTS) scavenging property, reducing power and metal ion chelating ability.

Radical scavenging capacity

The scavenging capacity of free radicals (OH•, O_2_^•−^, DPPH^•^, and ABTS) by degraded polysaccharides has been documented in many studies. The OH• radical is one of the most unstable (half-life of 10^−10^s) and strongest oxidants among the ROS, randomly able to attack biomolecules, and cause oxidative damage to certain proteins, lipids, and nucleic acids [109,122]. Degraded polysaccharides have been found to show higher OH• radical scavenging capacity compared to the native (undegraded) polysaccharides. The OH• radical scavenging rate would be dependent on the concentration of degraded polysaccharides [109,122]. Wang et al. [30] reported an increase in radical scavenging rate when the concentration of degraded polysaccharide was increased from 0.15 mg/mL to 3 mg/mL. Similarly, Xu et al. [40] found that degraded polysaccharides of varying molecular weights exhibited different degrees of scavenging effects on the hydroxyl group, and this effect increased with decreasing molecular weight of polysaccharides. In addition, a polysaccharide with a molecular weight of 3.26 × 10^4^ KDa, degraded to 1.30 × 10^4^ and 9.62 × 10^3^ KDa was found to show an increased OH• radical scavenging rates from 66.89% before H_2_O_2_ degradation to 92.58% and 93.21% in the degraded polysaccharide respectively [40].

The O_2_^•−^ radical (half-life 10^−6^ s) is another key member of ROS, produced continuously in biological systems, and could either cause harm to cell components or serve as a precursor to other unstable reactive free radical species [40,100]. Just like in OH• radical, degraded polysaccharide showed higher O_2_^•−^ radical scavenging capacity than the native polysaccharide, and the scavenging capacity was also enhanced with increasing concentration of degraded polysaccharide. The O_2_^•−^ radical scavenging capacity of native polysaccharide (21%) and degraded polysaccharide (39%) at a concentration of 0.2 mg/mL increased significantly to 53.41% (native polysaccharide) and 62.12% (degraded polysaccharide) when the concentration was raised to 1.2 mg/mL [40]. Similarly, Wu et al. [104] also reported a significant increase in O_2_^•−^ radical scavenging capacity (rate), when the concentration was 0.2 mg/mL with a scavenging rate of 12% and 32% for native polysaccharide and degraded polysaccharide was increased to 1.2 mg/mL, resulting in an increased scavenging capacity of 20% and 66% for native polysaccharide and degraded polysaccharide, respectively.

Furthermore, DPPH^•^ and ABTS radicals are (stable) free radicals frequently employed in radical scavenging assays for evaluating antioxidant capacity (of relevant compound(s)). Similarly, studies have shown that degraded polysaccharides showed significant scavenging capacity on DPPH^•^ and ABTS radicals, over the native polysaccharide [25,40,104,123] and the degree of the radical scavenging capacity of degraded polysaccharide on DPPH^•^ and ABTS radical were concentration-dependent. This implies that the degraded polysaccharide can easily react with free radicals to terminate their radical chain reaction or donate electrons from the degraded polysaccharide’s free hydroxyl group to reduce the radical to a more stable form [124]. Studies have also shown that degraded polysaccharides have more hydroxyl groups than their native polysaccharide which is an indication of a more available reductive hydroxyl group, which could accept and eliminate the free radicals [40,125]. Importantly, an increase in concentration and lower molecular weight of degraded polysaccharides play a significant role in the radical scavenging capacity [104], thus higher antioxidant capacity.

Reducing power

The reducing power of a compound is a potential indicator of its antioxidant activity [40,126]. Degraded polysaccharides with low molecular weight have strong reducing power. The reducing power of a blackcurrant polysaccharide with an initial molecular weight of 3.26 × 10^4^ KDa degraded to a molecular weight of 9.62 × 10^3^ KDa was found to increase from 0.183 before degradation to 0.414 in the degraded polysaccharide [40]. Additionally, a positive correlation exists between the concentration of polysaccharides and reducing power. The reducing power of native polysaccharide (0.02) and degraded polysaccharide (0.3) at a concentration of 0.2 mg/mL increased strongly to 0.11 in native polysaccharide and 0.87 in degraded polysaccharide when the concentration was increased to 1.2 mg/mL. This implies that reducing power increases with a decrease in molecular weight and increasing concentration of polysaccharides. The reducing power of degraded polysaccharides comes from their ability to donate a hydrogen atom to destroy free radical chain reactions, thus exerting its antioxidant activity [40,104]. The increased reducing power in low molecular weight degraded polysaccharides is attributed to the higher number of reducing and non-reducing ends, compared to their native polysaccharide [109].

Metal ion chelating ability

Transition metals such as iron and copper are well known to enhance the production of free radicals which are precursors of oxidative stress [40]. They are regarded as stimulants/facilitators of lipid peroxidation (Fenton reaction) consequently yielding more reactive alkoxyl and peroxyl radicals [5]. Amongst the transition metals, iron is the most important pro-oxidant due to its high reactivity [5,127]. Besides being an essential mineral for normal body physiology, the overproduction (excess) of iron can result in cellular damage from oxidative stress induced by free radicals. As a result, reduction in the concentration of the catalyzing transition metal-mediated autoxidation through metal ion chelation is commonly considered as an antioxidant mechanism [40,128]. Polysaccharides can inhibit free radical production by chelating metal ions instead of scavenging them directly [123]. Further, Chun-hui et al. [129] isolated two polysaccharide fractions (GAPS-1 and SAPS-1) from *A. barbadensis* Miller that showed good metal ion chelating ability against ferrous ion. Similarly, a polysaccharide from the fern *Lygodium japonicum* was reported to also show strong metal ion chelating potential [130,131]. Polysaccharide’s metal ion chelating ability has been shown to serve as a secondary (preventive) antioxidant because of their redox potential reduction ability which stabilizes the oxidized form of the metal ions [132].

Furthermore, studies have shown that degraded polysaccharides exhibited significantly higher metal ion chelating ability on Fe^2+^ over the native polysaccharide [40], as the metal-chelating effects of the degraded polysaccharides depend on concentration. Polysaccharide with a molecular weight of 3.26 × 10^4^ KDa degraded to 1.30 × 10^4^ and 9.62 × 10^3^ KDa was found to show an increased metal chelating ability from 5.0% before H_2_O_2_ degradation to 22.28% and 23.32 in the degraded polysaccharide respectively when concentration was increased from 0.2–1.2 mg/mL [40]. Specifically, the efficiency of polysaccharides in chelating metal ions is dependent on its structural configuration especially for functional groups of –OH, –SH, –COOH, –PO_3_H_2_, –C=O, –NR_2_, –S–, and –O– [123,133]. This was corroborated by Fan et al. [134] who confirmed that the chelating effect of degraded polysaccharides from leaves of *Ilex latifolia* Thunb. was partly due to the presence of functional groups such as the carboxyl group in the polysaccharide structure. Similarly, the presence of hydroxyl groups and the number of hydroxyl group on a polysaccharide structure have been found to influence the chelating affinity for the ferrous ion [123,135]. In line with polysaccharide’s ability to act as a metal ion chelator, its chelation mechanism is owed to complexes formed between the metal ion and aforementioned functional groups on the polysaccharide structure [136]. Polysaccharides having exclusively hydroxyl groups in equatorial position (for example, glucose molecule) do not seem to form complexes with metal ions, rather a solely favorable arrangement or configuration of two or three hydroxyl groups allows for the formation of metal ion complexes [137]. This implies that structural configuration and the arrangement of functional group (OH group) on a polysaccharide influences the formation of metal ion complex, which is the stable form of the oxidized metal ion.

### 4.3. The Reaction Mechanism Involving Polysaccharides’ Antioxidant Capacity on Free Radicals: A Primer

Polysaccharide has two mechanisms of degrading the OH• radical: (a) Removal of already generated OH• radicals through hydrogen electron transfer or single electron transfer [30,81]; and (b) inhibition of OH• radical formation [30,138]. Figure 7 shows the proposed reaction mechanism of OH• on a polysaccharide chain. Firstly, OH• radicals attack polysaccharides at different carbon positions and only a few of the polysaccharide backbones are influenced by these attacks, while oxo groups are simultaneously introduced into the polysaccharide chain, without altering the main chain. An OH• radical can abstract a hydrogen atom of the C-H chain of polysaccharides to form water, while on the other hand, a carbon-centred radical (carbon-free radical) is formed. In aerobic conditions, the carbon-free radical is oxidized to peroxy radicals and subsequently, reacts with hydroperoxyl radical (HO_2_•, often referred to as superoxide anion radical, O_2_^•−^ in its ionized state) to produce ketone, lactone, or aldehyde [139,140].

The introduction of oxo group (reactions c4 and d4) is a result of polysaccharide chain scission caused by the elimination of hydrogen from C-1 or C-4 position on a pyranose ring. Conversely, when hydrogen is abstracted from the C-5 position, there is no occurrence of polysaccharide chain scission, rather a more labile ester is formed (reaction a4). Additionally, when hydrogen is abstracted from the C-3, C-2, or C-6 positions, there is also no scission of polysaccharide chain but a relatively stable oxo group creating a glycosulose residue would be formed (reaction b3) [140]. Another mechanism of polysaccharides to exhibit their antioxidant capacity is the ability to inhibit free radical formation by chelating transition metal ions, especially Fe^2+^ that catalyse the formation of highly reactive OH• radical from O_2_^•−^ and H_2_O_2_ (Fenton reaction) [25,135,140].

### 4.4. Some Examples of Antioxidative Polysaccharides and Their Sources

Table 1 shows some examples of antioxidative polysaccharides and their sources. Out of 46 examples of antioxidative polysaccharides total presented herein, two (2) that include *Hyriopsis cumingii* and *Misgurnus anguillicaudatus* were of animal [25,141,142], four (4) that include *Bacillus coagulans*, *Edwardsiella tarda*, *Paenibacillus polymyxa*, and *Streptomyces virginia* were of bacterial [25,143,144,145,146,147,148,149,150,151], seven (7) that include *Bifurcaria bifurcata*, *Enteromorpha linza*, *Fucus vesiculosus*, *Laminaria angustata*, *L. japonica*, *Padina gymnospora*, and *Phaeophyceae spp* were of algae [25,36,152,153,154,155,156,157,158,159,160], fifteen (15) that include *Astragalus mongholicus*, *Auricularia auricula*, *Azadirachta indica* leaves, *Chuanminshen violaceum*, *Dendrobium huoshanense*, *Litchi chinensis* Sonn., *Longan*, *Lycium barbarum*, *Persimmon*, *Portulaca oleracea* L., pumpkin, *Astragali radix*, *Ribes nigrum* L. (blackcurrant fruits), *Scutellaria barbata* D. Don, and *Tremella fuciformis* were of plant [34,42,43,123,161,162,163,164,165,166,167,168,169,170,171,172,173,174,175,176,177,178,179,180,181,182,183,184], as well as eighteen (18) that include *Athyrium multidentatum*, *Calocybe indica* var, *Cordyceps gunni*, *Coriolus versicolor*, *Dictyophora indusiata*, *Flammulina spp*, *Ganoderma atrum*, *G. lucidum*, *Grifola frondosa*, *Lachnum*, *Lentinus edodes*, *Lentinus polychrous*, *Pleurotus abalonus*, *P. cornucopiae*, *P. eryngii*, *P. florida*, *P. nebrodensis*, and *P. ostreatus* were of fungal [15,25,38,44,117,131,185,186,187,188,189,190,191,192,193,194,195,196,197,198,199,200,201,202,203] origin.

From Table 2, we can see that the various sources, from the plant, animal, to bacteria and algae, gave various polysaccharide specificities. We can also see that within the same sources, aspects of chain-linkages here and there may be similar. Additionally, there could be some polysaccharide types in common. Nonetheless, several antioxidative polysaccharides discovered from plants, animals, fungi, bacteria, and algae sources [25] have been reported to exhibit antioxidant properties in in vitro and in vivo systems [144,204,205,206,207,208]. The antioxidative polysaccharides in these studies were found to possess reducing power, radical scavenging ability, and metal ion chelating property [25]. It is important to reiterate here that there are some other natural polysaccharides reported to exert biological and antioxidant effect that include, but are not limited to, white jelly mushroom (*Tremella fuciformis*) which is also known as ‘snow fungus’ [34], oyster mushroom (*Pleurotus* spp.) [209], blackcurrant (*Ribes nigrum* L.) [40], and polysaccharides from *Auricularia auricular, Radix astragali*, *Ganoderma lucidum* [34], *Athyrium multidentatum*, *Scutellaria barbata* D. Don, *Astragalus mongholicus*, *Litchi chinensis* Sonn., *Lycium barbarum*, *Lentinus edodes* and *Cordyceps sinensis* [25].

## 5. Concluding Remarks and Future Prospects

Polysaccharide degradation by the free radical-catalysed system offered by H_2_O_2_ is indeed effective in improving the bioactivity, especially the antioxidant activity. The current review has highlighted how H_2_O_2_ singly or in combination with other techniques can degrade natural polysaccharides. How the effects of polysaccharide degradation would influence the structural characteristics, physicochemical properties, and bioactivities (antioxidant capacity) of degraded polysaccharides has also been enumerated in this review. The free radical-catalysed polysaccharide degradation method offered by H_2_O_2_ is frequently employed because of its mild reaction conditions, high efficiency, inability to alter the integrity of the main chain structure of polysaccharides, and ease of OH• radical formation. Overall, the antioxidant capacity of polysaccharides is not dependent on a single factor but rather a combination of several related factors. Compared with the native polysaccharide, degraded polysaccharides exerts their antioxidant activities due to their low molecular weight, increased water solubility, decreased intrinsic viscosity, and modified structural characteristics.

Polysaccharide degradation comes with free radical scavenging capacity on OH• and O_2_^•−^ radicals, transition metal chelating ability, and reducing power through their hydrogen atom donating ability. However, there is a need to further understand what happens before the degradation, specific to the level of purity of polysaccharide with antioxidant properties, and this should be the direction of future literature syntheses/experimental programs to supplement existing information, given that other antioxidant compounds such as polyphenols may form part of the coextractives retained in the polysaccharide. Degraded polysaccharides may find wide application in the food and beverage industries as novel functional ingredients. The direction of future studies should focus on how polysaccharides and other food components interact within the food or beverage matrix, as this may either change, enhance or inhibit the antioxidant potential of the polysaccharide. The information this overview provides should prove useful, not only to biotechnologists, food scientists, and technologists, but also processors, as well as product developers of novel and functional food formulations.

## Figures and Tables

**Figure 1 foods-10-00699-f001:**
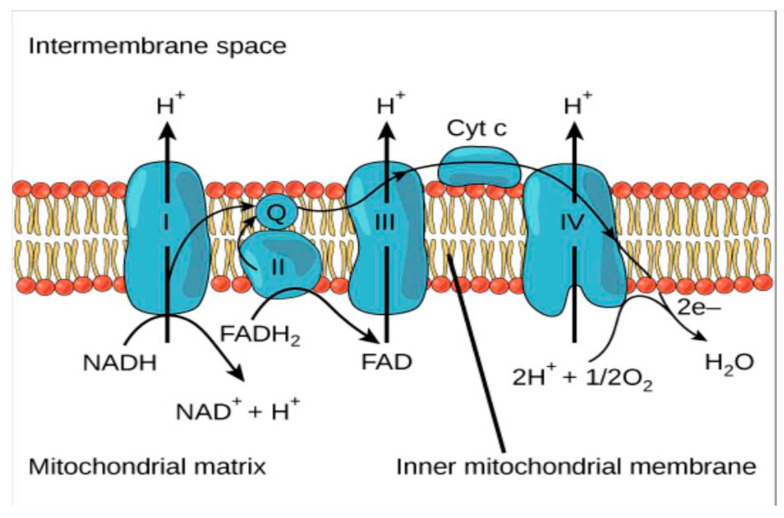
The electron transport chain is a series of electron transporters embedded in the inner mitochondrial membrane that shuttles electrons from NADH and FADH_2_ to molecular oxygen. In the process, protons are pumped from the mitochondrial matrix to the intermembrane space, and oxygen is reduced to form water (Source: Electron Transport Chain [50]).

**Figure 2 foods-10-00699-f002:**
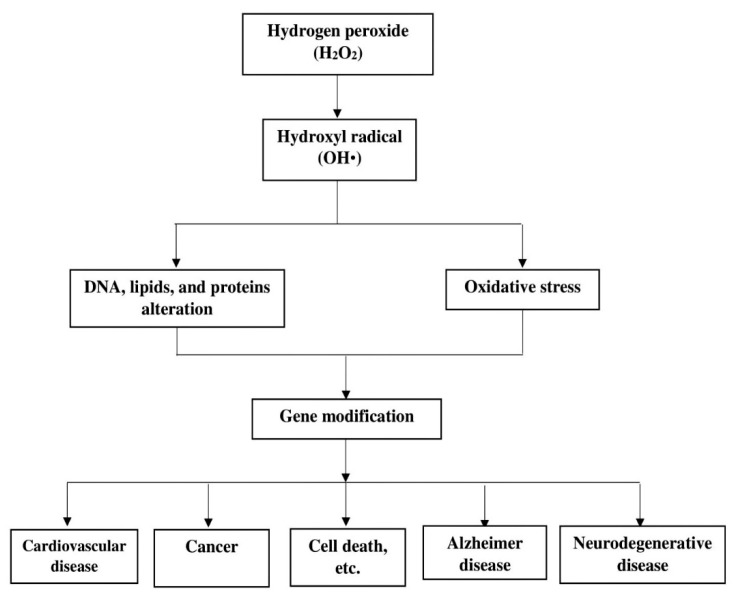
The role of hydrogen peroxide (H_2_O_2_) in cell dysfunction and transformation.

**Figure 3 foods-10-00699-f003:**
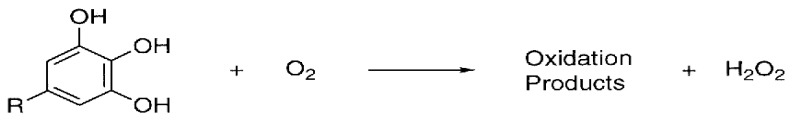
Proposed mechanism of oxidation of gallocatechin (Source: Nakayama et al. [61]).

**Figure 4 foods-10-00699-f004:**
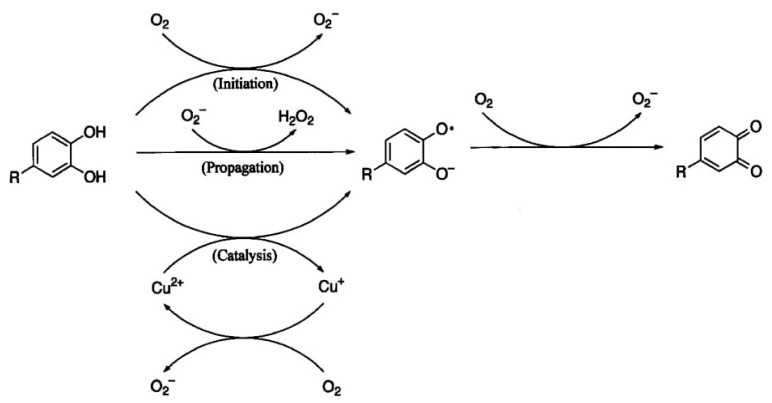
Production of hydrogen peroxide (H_2_O_2_) by polyphenols via autoxidation (Source: Akagawa et al. [62]).

**Figure 5 foods-10-00699-f005:**
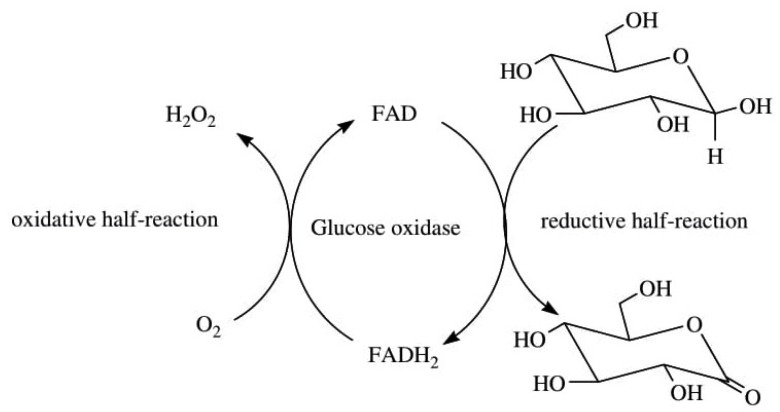
Schematic diagram for the oxidation of β-D-glucose, as catalysed by glucose oxidase (Source: Adányi et al. [85]).

**Figure 6 foods-10-00699-f006:**
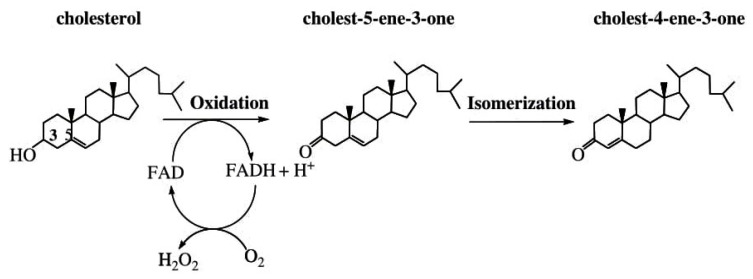
Schematic diagram of cholesterol oxidation and isomerization catalysed by cholesterol oxidase (Source: Adányi et al. [85]).

**Figure 7 foods-10-00699-f007:**
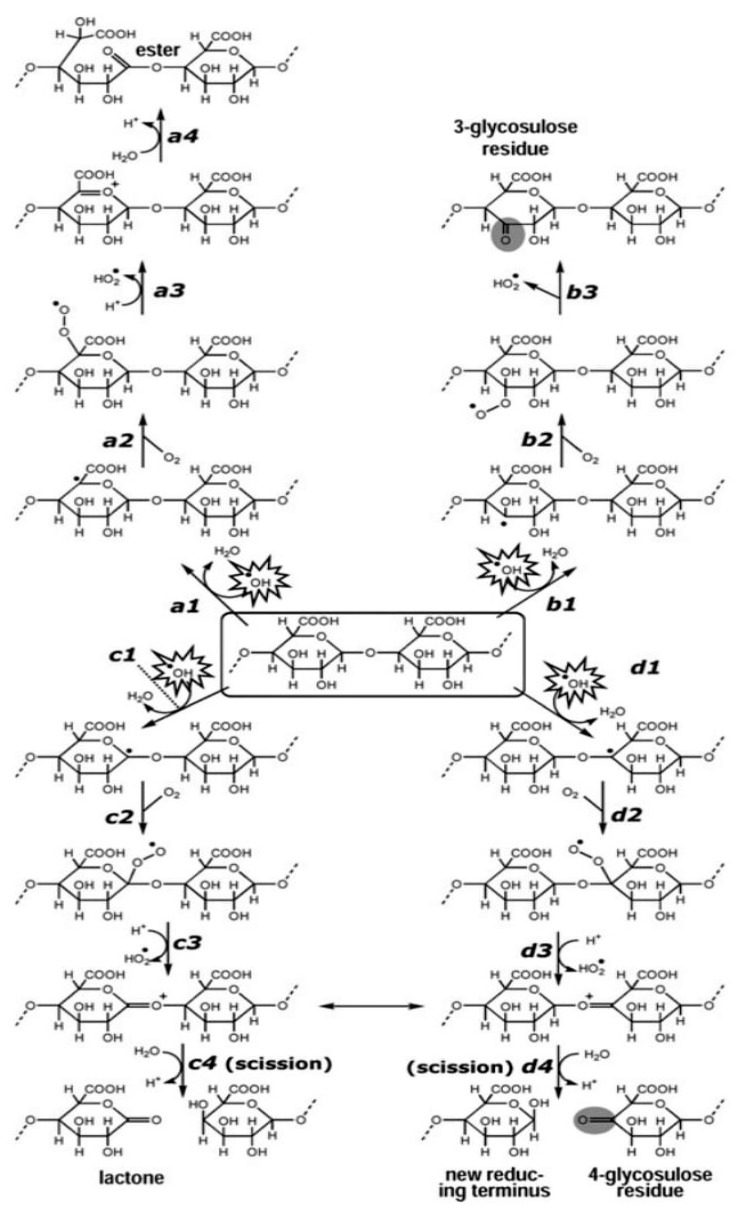
Proposed reaction mechanism of OH• on a polysaccharide chain. The diagram shows some of the proposed reaction mechanism of polysaccharide (homogalacturonan) exerting its antioxidant capacity on OH•, where the short lines (- - -) depicts the polysaccharide chain continuation. This figure presents the predicted reactions occurring after •OH abstracts a hydrogen atom from C-1 (reactions c1–c4), C-3 (reactions b1–b3), C-4 (reactions d1–d4) or C-5 (reactions a1–a4). Abstraction of hydrogen from C-2 (not shown) is expected to give products directly comparable with those shown for C-3. (Source: Vreeburg et al. [140]).

**Table 1 foods-10-00699-t001:** Some key situations associated with the formation of H_2_O_2_.

Location	In Vitro	Reference(s)	In Vivo	Reference(s)
**Source/site**	Beverages (e.g., green tea, black tea, wine, beer, coffee)	Grzesik et al. [63]	Mitochondria	Lennicke et al. [15]; Baveris and Cadenas [86]; Wong et al. [87]; Treberg et al. [88]
**Trigger agent**	Certain phenolic compounds (e.g., tea catechins, (+)-catechin, and gallic acid)	Wee et al. [67]; Grzesik et al. [63]	Escaped electron (e^−^) in the electron transport chain	Phaniendra et al. [7]; Druck et al. [89]; Maddu [2]
**Production process**	Auto-oxidation	Erickson [26]; Kim et al. [65]; Bensid et al. [27]	Cellular oxidative metabolism	Patel et al. [6]; Bottje [90]; Fang et al. [91]
**Factors responsible for H_2_O_2_ formation**	Content and composition of phenolic compounds; Structure of phenolic compounds; Oxygen; Metal ions; Temperature; pH; Incubation time.	Akagawa et al. [62]; Grzesik et al. [63]; Bopitiya et al. [59]	The action of certain enzymes such as superoxide dismutase (SOD), xanthine oxidase, monoamine oxidase, and glycollate oxidase; Type of ingested beverage such as green tea, black tea, and/or coffee; Available oxygen; The joint action of in vivo dismutation of superoxide radical and in vitro auto-oxidation of phenolic compounds	Fang et al. [91]; Grzesik et al. [63]; Bopitiya et al. [59]

**Table 2 foods-10-00699-t002:** Antioxidative polysaccharides, with corresponding sources, and polysaccharides specifics.

Antioxidative Polysaccharide	Sources	Polysaccharides Specifics	References
*Astragali radix*	Plant	From *Astragali radix,* isolated two glacans (AG-1, identified as as α-(1,4) and α-(1,6) glucan (5:2); and AG-2 identified as α-(1,4) glucan) and two heterosaccharides (AH-1, identified as acidic polysaccharide, composed of hexouronic acid, glucose, rhamnose, and arabinose in the ratio of 1:0.04:0.02:0.0; and AH-2, identified as glucose and arabinose in the ratio of 1:0.15) from an Astragali radix polysaccharides	Liu et al. [169]; Mckenna et al. [173]
*Astragalus mongholicus*	Plant	Polysaccharides from *A. mongholicus,* named as APS-I and APS-II, comprised backbone (1,3)-β-d-glucopyranosyl (Glcp) residues, with bioactivities closely related to their chemical composition, configuration, molecular weight, and physical properties	Yin et al. [182]; Zhu et al. [210]
*Litchi chinensis* Sonn.	Plant	A novel polysaccharide (LCP50W) from pulp tissues of *Litchi chinensis* had main chain consist of (1→3)-linked β-l-rhamnopyranosyl, (1→6)-linked α-d-glucopyranosyl, and (1→2,6)-linked α-d-glucopyranosyl residues, which branched at O-6. The three branches consisted of (1→2)-linked α-l-rhamnopyranosyl, (1→3)-linked α-d-galactopyranosyl, and (1→3)-linked α-l-mannopyranosyl residues, terminated with (1→)-linked α-l-arabinopyranosyl residues, respectively	Jing et al. [166]; Yin et al. [182]
*Lycium barbarum*	Plant	Raw and Purified *Lycium barbarum* polysaccharides (LBP) would contain sugars such as Rha, Gal, Glc, Ara, Man, Xyl, with molar ratio of 4.22, 2.43, 1.38, 1, 0.95, and 0.38, respectively	Luo et al. [172]; Yin et al. [182]
*Scutellaria barbata* D. Don	Plant	From the whole plant of *Scutellaria barbata* D. Don, polysaccharide, SBPW3, showed a molecular weight of 10.2 kDa and composed of arabinose (25.68%), galactose (27.72%), glucose (20.59%), mannose (12.56%), rhamnose (2.51%), and xylose (10.94%)	Li et al. [168]; Yin et al. [182]
*Ribes nigrum* L.	Plant	Polysaccharides from blackcurrant (*Ribes nigrum* L.) (BCP), showed two low-molecular-weight polysaccharides (DBCP-1, DBCP-2) with same monosaccharide units with resembling glycosidic linkage patterns	Xu et al. [179]; Yin et al. [182]
*Tremella fuciformis*	Plant	*Tremella fuciformis* polysaccharide (catechin-g-TPS) with molecular weight of 5.82 × 10^5^ Da, given by polysaccharide fractions (F-1 and F-2)	Liu et al. [170]; Wang et al. [123]; Zhang et al. [34]
*Auricularia auricula*	Plant	- Fruit bodies of *Auricularia auricula* were comprised of Glucans A, C, and E, mainly backbone chain of Beta (1-3)-d-glucose residues, with various branched groups. Specifically, polysaccharides D and B would have residues of d-xylose, d-mannose, d-galactose, d-glucose, and d-glucuronic acid. - Two polysaccharide fractions (AAPF, AAPP) purified from the fruiting body of *A. auricula*, showed AAPF with five monosaccharides, including glucose, rhamnose, arabinose, mannose, and galactose with a molar ratio of 16.74:1.0:1.18:1.0:1.0; and AAPP with four monosaccharides, namely arabinose, mannose, galactose, and xylose with the molar ratio of 15.59:1.52:4.76:1.0	Zhang et al. [183]; Khaskheli et al. [167]; Nguyen et al. [174]; Zhang et al. [34]
*Chuanminshen violaceum*	Plant	- *Chuanminshen violaceum* polysaccharides (CVPS) comprise d-carubinose and d-glucose, in the ratio of 1: 16.2. The average molecular and number-average molecular weight CVPS of 9.7632 × 10^5^ Da and 5.2270 × 10^4^ Da respectively. - Another study showed Chuanminshen violaceum polysaccharides (CVPs) with neutral polysaccharides, mainly composed of glucose and galactose, with molecular weights ranging between 233.69 and 11.02 kDa.	Dong et al. [164]; Song et al. [176,178]
*Dendrobium huishanense*	Plant	With molecular weights range 1.16 × 10^5^ to 2.17 × 10^5^ Da, with main monosaccharide compositions that include Man and Glc, with glycosidic linkages β-1,4-Man*p* and β-1,4-Glc*p*, and substituted with acetyl groups at O-2 and O-3 of 1,4-linked Man*p*	Deng et al. [163]; Qian et al. [175]
*Persimmon*	Plant	Sulphated modification, polysaccharide was obtained from fresh (*Diospyros kaki* L.) persimmon fruit (PFP), Three sulphated derivatives of PFP, known as PFP-SI, PFP-SII and PFP-SIII, were obtained, with average molecular weights of 53, 51 and 48 kDa, respectively,	Lu et al. [171]; Zhang et al. [184]
*Longan*	Plant	Longan polysaccharides (LPSs), extracted from (logan) pulp, showed β-type acidic heteropolysaccharides with pyran group, wherein LPS-N had glucose/xylose molar ratio of 1.9:1, with LPS-A1 having rhamnose, xylose, arabinose and galactose molar ratio of 1:1.64:4.33:2.28, and LPS-A2 having only rhamnose	Jiang et al. [165]; Yang et al. [181]
*Azadirachta indica* leaves	Plant	A native polysaccharide of *Azadirachta indica* (leaves) obtained an apparent molecular mass of 80 kDa, constituting (1→5)-/(1→3,5)-linked α-l-arabinosyl, (1→3)-/(1→6)-/(1→3,6)-linked β-d-galactosyl, and terminal-rhamnosyl and α-l-arabinosyl residues	Saha et al. [42]
*Portulaca oleracea* L.	Plant	A yield of 6.45% of dried raw material, *P. oleracea* showed water-soluble crude polysaccharides (DCPOP), 92% carbohydrate, free of proteins and with average molecular weight of 24.6 kDa	Chen et al. [161]; Shen et al. [42]
Pumpkin	Plant	Potentially containing acetyl groups, the pumpkin polysaccharide showed α/β-glycosidic bond linkage. It is considered a heteropolysaccharide, composed of six monosaccharides, namely arabinose, galactose, glucose, glucuronic acid, rhamnose, and xylose	Chen and Huang [162]; Song et al. [177]
*Ganoderma lucidum*	Fungi	Polysaccharides of *Ganoderma lucidum* consisted primarily of glucose and galactose, small amounts of mannose, rhamnose and fucose, and minor amount of glucuronic acid	Wang et al. [25]; Xu et al. [197]; Zhao et al. [201]
*Athyrium multidentatum*	Fungi	Five polysaccharide fractions (PS-1, PS-2, PS-3, PS-4, and PS-5) isolated from *Athyrium multidentatum* (Doll.) Ching, was heteropolysaccharides with different molecular weights and monosaccharide compositions	Jing et al. [38]; Wang et al. [25]
*Flammulina spp*	Fungi	Flammulina polysaccharide comprise glucan, which could have a mix of some other fractions, like fucosan, galactose, glycan, mannan, and xylan	Wang et al. [25]; Xin et al. [196]
*Coriolus versicolor*	Fungi	Polysaccharide of *Coriolus versicolor* contains extracellular polysaccharide (EPS) mainly β-1, 3/β-1, 6-linked d-glucose molecules, predominantly comprising glucose and small amounts of arabinose, galactose, mannose, and xylose	Wang et al. [194]; Wang et al. [25]
*Pleurotus nebrodensis*	Fungi	Polysaccharide (PNPS) from the fruiting body of *Pleurotus nebrodensis* contained mainly glucose, with a small percentage of galactose and mannose	Gao et al. [203]; Wang et al. [25]
*Pleurotus abalunes*	Fungi	Polysaccharides of *Pleurotus abalonus* is heteropolysaccharide, comprise d-mannose, d-ribose, l-rhamnose, d-glucuronic acid, d-glucose and d-galactose, having corresponding mole percentages 3.4%, 1.1%, 1.9%, 1.4%, 87.9% and 4.4%, respectively	Ren et al. [191]; Wang et al. [25]
*Pleurotus eryngii*	Fungi	Polysaccharides from *P. eryngii* comprise mainly of β-(1→3)-glucans with β-(1→6) branches	Jung et al. [189]; Wang et al. [25]
*Pleurotus corncopiae*	Fungi	Polysaccharides isolated from *Pleurotus cornucopiae* composed of arabinose, glucose, mannose and xylose	Juyi et al. [190]; Wang et al. [25]
*Pleurotus florida*	Fungi	*P. florida* consists of three different polysaccharides, which include (1→3)-, (1→6)-branched glucan, and (1→6)-α-glucan	Rout et al. [192]; Wang et al. [25]
*Calocybe gambosa*	Fungi	-Three polysaccharides were isolated from the fruiting bodies *Calocybe gambosa* mushroom, composed of glucose, and methylation analysis, showed the units were (1→4),(1→6)-linked with a degree of branching (DB) of 4%	Villares [193]; Wang et al. [25]
*Lentinus edodes*	Fungi	*L. edodes* can be divided into three layers: (1) the outside layer has heteropolysaccharide and β-(1→3)-glucan with β-(1→6) branches; (2) middle layer has mainly β-(1→6)-glucan with a small number of β-(1→3) branches; (3) inner layer being complex chitin, β-glucan, with small amount of acid polymer	Xu et al. [198]; Wang et al. [25]
*Lentinus polyschrous* Lev.	Fungi	The major monosaccharides of *Lentinus polychrous* include d-galactose, d-glucose, and d-mannose at different proportions, fucose, and xylose	Ayimbila and Keawsompong [185]; Wang et al. [25]
*Pleurotus ostreatus*	Fungi	Polysaccharide isolated from *Pleurotus ostreatus* is heteropolysaccharide, composed of galactose, glucose, mannose, rhamnose, and xylose	Vamanu [15]; Wang et al. [25]; Zhang et al. [200]
*Cordyceps gunni*	Fungi	The polysaccharide of *Cordyceps gunnii* was the *α*-polysaccharide type, with a glycosidic bond, mainly composed of d-mannose, d-glucose, and d-galactose	Zhu et al. [202]; Zhu et al. [45]
*Grifola frondosa*	Fungi	*G. frondosa* polysaccharides contain glucose, galactose, mannose, fucose, and ribose, with high amounts of (1→3,1→6)-β-d-glucans, which would account for 13.2% of water-soluble polysaccharides	Bae et al. [117]; He et al. [131]
*Lachnum*	Fungi	Polysaccharide of *Lachnum* YM261(LEPS-1) is considered homogenous with a molecular weight of 21,670 Da, with its glucan linked by β-(1→3)-d-pyran glycosidic bond	Wu et al. [195]; Ye et al. [199]
*Dictyophora indusiata*	Fungi	Polysaccharide (DIP) extracted from *Dictyophora indusiata* indicated a specific polysaccharide of (1→3)- β-d-glucan with (1→6)-β-glucopyranoside side chains	Deng et al. [187]; Deng et al. [188]
*Ganoderma atrum*	Fungi	Polysaccharide fraction from *Ganoderma atrum* composed of glucose (Glc), mannose (Man), galactose (Gal) and galacturonic acid (GalA) in molar ratio of 4.91:1:1.28:0.71	Chen et al. [186]; Zhang et al. [200]
*Paenibacillus polymyxa*	Bacteria	- *Paenibacillus polymyxa* bioactive compounds like exopolysaccharides, for example, d-glucuronic acid, and polysaccharide beads - *Paenibacillus polymyxa* was shown to comprise mannose, galactose, and glucose in a ratio of 1.23:1.14:1	Daud et al. [143]; Raza et al. [151]; Wang et al. [25]
*Bacillus coagulans*	Bacteria	- *Bacillus coagulans* polysaccharide could have with monosaccharides like galactose, glucose, lactose, mannose, raffinose, rhamnose, and xylose - Exopolysaccharide (EPS) from *Bacillus coagulans*, comprise galactose, mannose, fucose, glucose, and glucosamine	Kodali et al. [149]; Ramanathan et al. [150]; Wang et al. [25];
*Edwardsiella tarda*	Bacteria	- *E. tarda* polysaccharide comprise glucosamine, a fucose, a mannose and a galactosamine unit; - Another *E. tarda* polysaccharide comprise d-glucuronic acid with l-alanine, branched hexasaccharide repeating unit. - branch points situated at C-2 and C-6 positions of the (1→3)-linked mannose residues, whereas the side chains composed of (1→2)-linked and (1→)-linked mannose residues	Guo et al. [144]; Katzenellenbogen et al. [147,148]; Wang et al. [25]
*Streptomyces virginia*	Bacteria	-A purified polysaccharide *Streptomyces virginia* (HO3 type) comprised mannose, glucose and galactose, in a 2:1:1 proportion, with average apparent molecular weight of 3.76 × 10^4^ Da	He et al. [145]
*Hyriopsis cumingii*	Animal	Polysaccharides of *Hyriopsis cumingii* have pyranose rings, with glycosyl residues, either linked by α- and β-configuration glycosidic bonds	Qiao et al. [141]; Wang et al. [25]
*Misgurnus anguillicaudatus*	Animal	*Misgurnus anguillicaudatus,* polysaccharide (MAP) or misgurnan, comprise the major structural monomers identified included d-galactose, l-fucose and d-mannose, and monoses that link each other by α-1,3 bonds	Qin et al. [142]; Wang et al. [25]
*Laminaria japonica*	Algae	sulfated polysaccharide (L-A) has high galactose content but low fucose residues proportion	Wang et al. [25]; Xu et al. [160]
*Enteromorpha linza*	Algae	*Enteromorpha linza* polysaccharides contain large amount of sulfated polysaccharides, composed mainly of galactose, mannose, glucuronic acid, glucose, rhamnose, and xylose	Wang et al. [25]; Zhang et al. [36]
*Bifurcaria bifurcata*	Algae	Glucose and *myo*-inositol have been detected in *Bifurcaria bifurcata*	Mian and Percival [152]; Soukaina et al. [156]
*Phaeophyceae spp*	Algae	Sulfated polysaccharides (SPs) found in *Phaeophyceae spp* can include laminarans, which comprise repeating dimeric units of λ-carrageenan	Wang et al. [25]; Wijesekara et al. [159]
*Fucus vesiculosus* Linnaeus	Algae	- The main neutral sugars of *Fucus vesiculosus* include fucose, glucose, galactose, and xylose - Sulphated polysaccharides of *Fucus vesiculosus* has a heterogeneous structure	Rodriguez-Jasso et al. [153]; Rupérez, Ahrazem, and Antonio Leal [154]; Wang et al. [25]
*Padina gymnospora* Sonder	Algae	Sulfated polysaccharide (SP) isolated from *Padina gymnospora* contains 29.4 ± 0.35% of sulfate, 27 ± 0.11% of fucose, 0.05 ± 0.12% of protein	Vasantharaja et al. [158]; Wang et al. [25]
*Laminaria angustata*	Algae	Polysaccharides from *Laminaria angustata* obtained via fucoidan -led macrophage activation produced chemical composition ratio of l-fucose:d-galactose:d-glucose:d-xylose:uronic acid:sulfate = 1.00:0.54:0.08:0.08:0.64:0.87	Saha et al. [155]; Teruya et al. [157]

## Data Availability

Data sharing is not applicable.

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
