# Peer review of "Hydrogen Peroxide Effects on Natural-Sourced Polysacchrides: Free Radical Formation/Production, Degradation Process, and Reaction Mechanism—A Critical Synopsis"

_foods, 2021, doi:10.3390/foods10040699_

Round 1

Reviewer 1 Report

This manuscript by Ofoedu et al. nicely put up an in-depth review on the hydrogen peroxide-catalyzed free radical degradation of polysaccharides. I think this review will be of interest to the Food Science and related broader scientific community. However, I have a few minor concerns which I listed below.

  1. The main focus of this review must be restricted on the ROS mediated degradation of the polysaccharide. A major part of the review discussed the production and the factors affecting the production of H2O2, which in my opinion dilutes the focus. Please keep those sections concise.  
  2. Main text and Fig 2: For hydroxyl or any other radical use proper radical sign () and not an asterisk. 
  3. Fig 2: Correction - neurodegenerative diseases
  4. Fig 2 legend: Correction- H2O2
  5. Page 22/ line 921 : Where is the Table ?
  6. Most of the figures with the chemical structures are of very low resolution. Please adjust them to high resolution.

Author Response

RESPONSE TO REVIEWER`S 1 COMMENTS

REVIEWER 1

Comments and Suggestions for Authors

This manuscript by Ofoedu et al. nicely put up an in-depth review on the hydrogen peroxide-catalyzed free radical degradation of polysaccharides. I think this review will be of interest to the Food Science and related broader scientific community. However, I have a few minor concerns which I listed below.

  1. The main focus of this review must be restricted on the ROS mediated degradation of the polysaccharide. A major part of the review discussed the production and the factors affecting the production of H2O2, which in my opinion dilutes the focus. Please keep those sections concise.  

Reply: Thank you very much for your kind comment. We appreciate very much that you made time available despite your occupied schedules to read well through our review manuscript. Importantly, we now see your view, and why you feel this review must be restricted to the ROS-mediated degradation of the polysaccharide. We  believe your opinion emanated from the previous title, ‘Hydrogen peroxide-catalysed free radical degradation of polysaccharides from natural sources: A critical synopsis’. With the previous title, you have a valid point that the focus  had to be restricted on the ROS mediated degradation of the polysaccharide.

However, this was not our goal in this review paper. Thank you for identifying this very important observation. Indeed, it is clear that the previous title was not appropriate because it did not capture our review manuscript’s focus/objective.

Firstly, thanks to your very important observation, we have now restructured the title to  be more appropriately capture the objective of the review.  The current title now reads: “Hydrogen Peroxide Effects on Natural-sourced Polysaccharides: Free Radical Formation/Production, Degradation Process, and Reaction Mechanism - A Critical Synopsis” .

Secondly, please permit us to reiterate here the key aim of this review paper, so as to help persuade you to agree with us, in this. You see,  the key aim of this current critical synopsis was to articulate the H2O2 effects on naturally sourced polysaccharides. So, there are two key domains, on one side is “H2O2 effects “ , and on the other side is “naturally sourced polysaccharides”.

Thirdly, the next step is the focus of this review paper. Specifically, if you will please permit,  this critical synopsis is focused on the free radical formation/production, polysaccharide degradation processes with H2O2, the effects of polysaccharide degradation on the structural characteristics; physicochemical properties; and bioactivities; in addition to the antioxidant capability. The degradation mechanisms involving polysaccharide’s antioxidative property; with some examples and their respective sources were tersely summarised.

You will surely agree with us now, both title and content of the work go along well in unity. The title now provides an appropriately clear snapshot of the review’s content consistent with its sub-headings, contents, contexts, etc.

Thank you for highlighting this, because it has allowed us to critically evaluate this entire (now ‘revised’) manuscript.

Please, we have checked and re-checked through the sub-sections again, and made efforts to keep it concise, as you have thankfully advised.

  1. Main text and Fig 2: For hydroxyl or any other radical use proper radical sign () and not an asterisk. 

Reply: All hydroxyl or any other radical sign marked as asterisk (*) has been changed to proper radical sign (•). 

  1. Fig 2: Correction - neurodegenerative diseases

Reply: Thank you for the observation. The correction of neurodegenerative diseases has been implemented in this revised manuscript.

  1. Fig 2 legend: Correction- H2O2

Reply: Thank you for the observation. The correction has been implemented in this revised manuscript

  1. Page 22/ line 921: Where is the Table?

Reply: Thank you. The Table has been inserted. It now exist as Table 2.

  1. Most of the figures with the chemical structures are of very low resolution. Please adjust them to high resolution.

Reply: Thank you for the observation. All low resolution figures have been replaced with clearer and higher resolution ones.

Reviewer 2 Report

The topic is suitable for publication in Foods. The work is well written and organized. In my opinion it could be published in Foods. However, there are some issues that should be addressed before acceptance. The list of references must be updated. There was a lot done in this field in the last 3 years and the paper barely refers to works published in those years. Making reference to recent work in the field is particularly key to highlight the current context of the present manuscript and to make it more comprehensive and to highlight the novelty to the readers as well as its assessment and contribution to the field. Please, address this request by adding new critical analysis and not by simply citing papers published in this subject in the mentioned years.

Author Response

RESPONSE TO REVIEWER`S 2 COMMENTS

REVIEWER 2

Comments and Suggestions for Authors

The topic is suitable for publication in Foods. The work is well written and organized. In my opinion it could be published in Foods. However, there are some issues that should be addressed before acceptance. The list of references must be updated. There was a lot done in this field in the last 3 years and the paper barely refers to works published in those years. Making reference to recent work in the field is particularly key to highlight the current context of the present manuscript and to make it more comprehensive and to highlight the novelty to the readers as well as its assessment and contribution to the field. Please, address this request by adding new critical analysis and not by simply citing papers published in this subject in the mentioned years.

Reply: Thank you very much for your kind comment. New literature syntheses with recent publication dates have been added in the revised manuscript.
